# Food insecurity and the nutritional health and well-being of women and children in high-income countries: protocol for a qualitative systematic review

Zoe Bell 🔟 , Steph Scott, Shelina Visram, Judith Rankin, Clare Bambra, Nicola Heslehurst

Population Health Sciences Institute, Faculty of Medical Sciences, Newcastle University, Newcastle upon Tyne, UK

**Correspondence to**
Zoe Bell; z.bell2@ncl.ac.uk

## ABSTRACT

**Introduction** Since the global financial crises of 2008, there has been a rise in the number of people experiencing food insecurity. The COVID-19 pandemic has exacerbated this. Many more are unable to afford or access food of sufficient quality and quantity to enable good health and well-being. Particularly vulnerable are mothers with young children, pregnant women and lone parents (the majority of whom are women). This review aims to understand experiences of food insecurity and how it affects women and children's nutritional health and well-being, focusing on experiences explicitly related to nutrition. Findings will help guide health policy and practice to support food-insecure women and children from high-income countries.

**Methods and analysis** A systematic review and meta-ethnography exploring (1) food-insecure women's own accounts of their nutritional health and (2) food-insecure household's accounts of their children's nutritional health. Six major databases (MEDLINE, Scopus, Web of Science, EMBASE, CINAHL and ASSIA), grey literature databases and relevant stakeholder websites will be searched from 1 January 2008 to 30 March 2021. Reference list and citation searches will supplement electronic database searches. Outcomes of interest are accounts of nutrition and nutritional health, including diet, food practices, infant feeding practices and physical and mental health. The review will follow the Preferred Reporting Items for Systematic Review and Meta-Analysis Protocol guidelines, but as this is a meta-ethnography it will adhere to eMERGe Reporting Guidance for synthesis and writing findings of the final report. Critical Appraisal Skills Programme qualitative checklist will assess the quality of studies. A meta-ethnographic analysis will be conducted for all included studies.

**Ethics and dissemination** As a qualitative systematic review, without primary data collection, ethical approval will not be required. Findings will be submitted for peer-reviewed publication.

**PROSPERO registration number** CRD42020214159.

## BACKGROUND

The terms food insecurity and food poverty both signify 'the inability to consume an

---

**Strengths and limitations of this study**

► Employing a rigorous international gold standard methodology (Preferred Reporting Items for Systematic Review and Meta-Analysis Protocols, PRISMA-P) to facilitate the development of this protocol and review conduct (PRISMA).

► Adherence to eMERGe Reporting Guidance for synthesising and writing up the findings of the review.

► Comprehensive broad search strategy, supplemented with reference and citation searches.

► A potential limitation of the review is that studies will be from different countries with different welfare states, social security, food aid and healthcare systems, which may impact on temporal relevance of findings.

---

adequate quality or sufficient quantity of food in socially acceptable ways, or the uncertainty that one will be able to do so'.[1] Accessing food in a socially acceptable way means people do not have to live off 'free' food redistributed by charity.[2] In high-income countries (HICs), food insecurity is generally not the result of supply failures but retrenchment of welfare states and diminishing rights to access healthy food.[3–6] The financial crises of 2008 meant economic downturn for all HICs, with countries responding differently to the crises. However, since 2008, poverty rates have increased and reports have consistently documented increased use of foodbanks, a proxy measure of food insecurity.[7 8] The COVID-19 pandemic has worsened food insecurity which was already an urgent public health issue.[9] Governments imposed, to varying degrees, lockdown measures that increased social isolation and confinement within the home to reduce the reproduction rate of the virus.[10] The effects are likely to be unequal

across society and are expected to lead to higher poverty rates, having profound effects on economically vulnerable families.[11] Families who were teetering on the edge of just managing may be pushed into food insecurity, thereby experiencing hunger, reduced food consumption, and creating an inability to secure food of sufficient quality and quantity to enable good health and participation in the society. Particularly vulnerable are families with young children, pregnant women and lone parents, the majority of which are women.[12]

It is well established that a healthy balanced diet is an important factor for health. Preconception health is strongly linked not only to pregnancy outcomes, but life course research evidences this period as crucial for health across generations,[13] widely recognised is the importance of nutrition during the first 1000 days of life (conception, pregnancy to 2 years of age).[14] As the diet of a young child is largely determined by the mother, understanding how food insecurity affects women's nutritional health and well-being is important. To date, most reviews exploring the effects of food insecurity on nutrition have reviewed quantitative studies.[15–18] Quantitative analyses reveal that individuals experiencing food insecurity are more likely to have a poorer quality diet than those who are food secure.[19] Food insecurity is adversely associated with dietary quality for adults; consuming fewer vegetables, fruits and dairy products than food secure adults and having lower intake of vitamins A and $B_6$, calcium, magnesium, iron and zinc.[16] For children, food insecurity is less consistently associated with lower dietary quality although there is substantial evidence of lower intake of fruits and vegetables.[16] Factors influencing the relationship between food insecurity and diet quality remain unclear.

Quantitative studies also report on nutrition-related health outcomes, food insecurity being associated with many non-communicable diseases.[15 20] A major public health concern is the robust association of food insecurity with obesity among children and adults in HICs.[15 20–23] Substantial risks for maternal and child health are associated with mothers who are overweight.[13] Explanations for the relationship between food insecurity and nutrition-related health remain unclear. Proposed theoretical explanations include sacrifice theory where mothers forego food for their child (supported by the strong relationship between food insecurity and obesity for mothers), substitution hypothesis whereby nutrient-dense, low-energy foods are substituted for cheaper, energy-dense, often higher fat and sugar-containing foods[24] and the insurance hypothesis associated with the cyclical nature of food insecurity and its disruption on metabolism.[25] Emotional distress has also been theorised as important in the relationship between food insecurity and nutrition-related health.[26 27]

Qualitative research can provide deeper insight into this area. A body of qualitative evidence is accumulating documenting women and children's experiences and perceptions of food insecurity in relation to their nutritional health and well-being. Food-insecure participants report altered diets via restrictive eating patterns, for example, reducing portion size, skipping meals or forgoing an entire day of food.[28–30] Food-insecure participants also report worsened health issues, such as stress, depression and weight gain, which have a knock-on negative effect on the health and well-being of dependent children.[28 31 32] Breast feeding is the healthiest method of infant feeding, yet a paradox exists where infants from lower income households are least likely to be breast fed, or for as long, despite arguably gaining most from the health and cost benefits it provides.[33 34] Reviewing qualitative research will help develop an understanding of the complexities around food insecurity and nutritional health and well-being.

Furthermore, despite most health research for children and young people being based on parents, caregivers or stakeholders' views, there is evidence that parents are not always aware of their children's experiences of food insecurity.[35] Studies have reported inconsistencies between parents and children's perspectives of the child's experience of food insecurity.[35] Indeed, evidence shows that children experience food insecurity in different ways to adults.[36–38] Systematically reviewing the literature for family unit's perspectives of the effects of food insecurity on children's nutritional health and well-being is important to explore the differences and gain a holistic view of its effect.

This review is part of a wider PhD project which explores the social determinants of food insecurity and its nutritional impact among women and children, focusing on the first 1000 days of life. The aim of this qualitative systematic review is to explore food-insecure women and children's experiences of nutritional health and well-being in HICs, within the context of the last 12 years. This review will further our understanding, inform health policy and practice, and result in recommendations for researchers and areas for further research.

## METHODS

Preliminary searches were carried out in September 2020 and the study was registered with PROSPERO[39] on 23 October 2020. The review started in November 2020 and is anticipated to take 11 months to complete. The review employed a rigorous international gold standard methodology (Preferred Reporting Items for Systematic Review and Meta-Analysis Protocols, PRISMA-P) to facilitate the development of this protocol and review conduct (PRISMA) (see online supplemental file A).

The review will use meta-ethnography, one of the most developed and structured methods to synthesise qualitative findings, developed by Noblit and Hare.[40] Meta-ethnography involves a seven-phase approach: (1) getting started; (2) deciding what is relevant to the initial interest; (3) reading the studies; (4) determining how the studies are related; (5) translating the studies into one another; (6) synthesising translations; (7) expressing the synthesis.

## Review questions

1. How do women perceive food insecurity to affect their nutrition and nutritional health and well-being?
2. How do food-insecure households perceive food insecurity to affect their child's nutrition and nutritional health and well-being?

## Objectives

The study objectives are:

1. To explore food-insecure women's experiences and accounts of their own nutritional health and well-being.
2. To explore food-insecure household's accounts of their children's nutritional health and well-being.

## Eligibility criteria

Studies will be selected according to the inclusion and exclusion criteria outlined in table 1. Eligibility criteria are outlined in accordance with the modified PICO tool,

**Table 1** Eligibility criteria for screening of studies

|  | Inclusion | Exclusion |
|---|---|---|
| Population (P) | Food-insecure women of childbearing age of all ethnicities (objective 1).<br><br>Food-insecure households (parents, primary caregivers, children) of all ages and ethnicities (objective 2). | Studies restricted to a specific type of population not directly related to women and children/wider population with clinical needs, which necessitates a specific diet (eg, studies in the context of people living with HIV, type 1 diabetes, etc).<br><br>Studies based on university campus with college students (unless in the context of also being a parent).<br><br>Perspectives of those outside the household family unit, that is, grandparents, healthcare professionals, teachers or food programme coordinators (objective 2). |
| Intervention (I) or exposure | Food insecurity.<br><br>Other terms used to describe food insecurity and included are, for example, food poverty, food deprivation, food insufficiency, hunger. | Food secure population groups.<br><br>Studies that were qualitative process evaluations of food insecurity-related interventions/services and focused on women and/or children. |
| Comparison (C) | Not applicable—systematic review of qualitative studies. | |
| Outcomes (O) | Experiences and accounts of the effect of food insecurity on nutrition and nutritional health and well-being. | Experiences and accounts not explicitly related to food. |
| Study type | Qualitative studies of any design including but not limited to: ethnography, interviews, focus groups, photo elicitation, visual techniques, phenomenology, grounded theory, case study, feminist research, action research.<br><br>Mixed methods studies.<br><br>Primary data sources from grey literature and relevant stakeholder websites. | Quantitative studies.<br>Reviews.<br>Expert opinion articles.<br>Editorials.<br>Policy documents.<br>Conference abstracts.<br><br>Qualitative research that reports no lay perspectives but has analysed text, that is, discourse analysis.<br><br>Grey literature that does not include primary qualitative data. |
| Study period | Published in the last 12 years (1 January 2008 to 2021).<br><br>Studies with data collected from 2008 onwards. | Literature published before 1 January 2008.<br><br>Studies reporting data only collected before 2008. |
| Setting | High-income countries (as per The Organisation for Economic Co-operation and Development, OECD, definition, see online supplemental file B). | Non-high-income countries. |
| Study reporting language | English. | |

PICOS, which is deemed appropriate for use in qualitative evidence reviews.[41 42] The PICO tool focuses on Population, Intervention, Comparison and Outcomes, whilst PICOS is modified to include Study design.

### Exposure

This review will explore the experiences and accounts of those who are food insecure. Food insecurity is defined as 'the inability to consume an adequate quality or sufficient quantity of food in socially acceptable ways, or the uncertainty that one will be able to do'.[1] Other terms used to describe food insecurity are included, for example, food poverty, food deprivation, food insufficiency and hunger. In this review, we will include those who are experiencing life on low income, in receipt of income benefit, those accessing food aid and those accessing food through public health programmes, for example, The Special Supplemental Nutrition Program (SNAP) for Women, Infants and Children (WIC) and Healthy Start vouchers.

### Outcomes

#### Primary

Studies will be included if they report experiences and accounts relating to the effect of food insecurity on nutrition and nutritional health and well-being. In the context of this review, nutrition outcomes can be reported as accounts of diet (quality and quantity of food, eating behaviour, eating pattern), food practices (ie, food acquisition, food preparation, organisation and storage of food in the house) and infant feeding practices (breast feeding, infant formula and complimentary feeding behaviour). Nutritional health and well-being outcomes for women and children include physical (eg, perspectives on their weight or growth and development of a child) and mental (eg, anxiety about household food running out).

Studies exploring food-insecure women and children's health more broadly will not be included unless the health effects are explicitly linked to nutrition. Examples of non-nutrition-related health effects of food insecurity include stress, anxiety or depression experienced in the context of being unable to pay the household bills or generally living in poverty where there is no mention of nutrition.

### Study exclusion

Studies will be excluded if they were published prior to 2008. The year 2008 was selected as a start date because this marks the beginning of the global financial crises, since which there has been a rise in the number of people experiencing food insecurity. Studies will be excluded if data collection took place prior to 2008. This is to capture the experiences of food insecurity postglobal financial crises. Only HICs will be included. See online supplemental file B for a list of HICs as per The Organsation for Economic Co-operation and Development (OECD) high-income economies definition.[43]

### Study design

Qualitative studies of any design will be included if they report experiences and accounts of food-insecure women

(objective 1) and/or experiences and accounts of household (parents, primary caregivers, children, objective 2) nutrition and nutritional health and well-being. Mixed methods studies will be included if qualitative data can be extracted independently from quantitative data. Primary data sources from grey literature-relevant stakeholder websites will be included if they report primary qualitative data relating to the review.

### Search strategy

We will search six bibliographic databases (MEDLINE, EMBASE, CINAHL, Scopus, Applied Social Science Index and Abstracts (ASSIA), Web of Science) from 1 January 2008 until 30 March 2021. Theses and dissertations will be searched using Open Access Theses and Dissertations (OATD). Grey literature will be searched using OpenGrey Europe (for information on Grey Literature in Europe) and Trove (links to Australian grey literature). Relevant stakeholder websites will be searched in March 2021 (see online supplemental file C). Only publications in English language will be included.

Search terms were identified from literature within the field, and the search strategy has been designed and piloted with an information scientist at Newcastle University. The strategy consists of four main concepts in accordance with the PICOS tool[42] (table 1). An example search strategy for Scopus can be found in online supplemental file D.

The descriptive titles of qualitative studies often lead to inappropriate indexing, posing challenges in finding relevant studies when searching bibliographic databases alone.[44] For this reason, we will screen included studies' reference lists for other studies published prior to the included study and use citation searching for studies citing the included study using Google Scholar.

### Study selection

Studies will be imported into EndNote V.X9.3.3[45] for deduplication, then imported into Rayyan,[46] an online program for systematic reviews. All titles and abstracts will be screened by ZB and a second reviewer (split between SS, SV and NH). Full texts will also be double screened. A pilot exercise screening 30 titles and abstracts will be carried out across the screening team (ZB, SS, SV) to calibrate and test the full-text review form. A third reviewer will assist to resolve any disagreements. Reasons for exclusion at the full-text stage will be recorded and a PRISMA flow chart will be used to report each stage of screening.

A standardised data extraction table will be created as a data extraction form and piloted using a subset of included studies. The following key information will be extracted from included studies: aim(s), country, time period, population characteristics (e.g., type of participant, age ranges, socioeconomic status (SES) measured in terms of income, education, occupational class), sample size, study methods, summary of the main themes with some exemplary direct quotes and context of the research. During this stage, we will actively look

for reporting on each study's context to explore in the analysis. This information will form a table of included studies. Data will be extracted by one reviewer (ZB).

## Risk of bias assessment

Quality of included studies will be assessed using the Critical Appraisal Skills Programme (CASP) qualitative checklist.[47] The checklist covers the primary focus of the paper, appropriateness of the study design, sample recruitment, methodology, analysis and generalisability. Quality appraisal will be undertaken by one reviewer (ZB), a sample of included studies will be double reviewed for quality appraisal to check agreement. CASP appraisals will be used to inform the data synthesis stage and provide an overview of the quality of included studies for context and inform discussion of the strengths and limitations of existing evidence.

## Data synthesis

Methods for review synthesis are to be determined through informed conversations of the nature of the evidence available, the review questions and purpose.[48] Meta-ethnography places studies side by side to see how key themes can be translated between studies while considering similarities and differences across varied contexts. This interpretive approach moves beyond describing or aggregating findings, instead aiming to 'synthesise understanding'.[40] If required, to 'synthesise understanding' we will contact the authors of studies to check interpretations of important points.[49] The integration of findings of multiple studies will enable the development of deeper insights into the understanding of food insecurity in the context of nutrition, which individual studies alone cannot provide.

Meta-ethnography synthesis will be conducted in seven steps using NVivo V.10 software.[50] Step 1, in-depth reading of included studies by three reviewers (ZB, SS and SV). Step 2, creation of study subsets and line-by-line coding and extracting of first and second order themes. The third step is to determine how studies are related. We will tabulate first order themes (interpretations) and second order themes (interpretation of interpretations) with grouped studies to create 'meta-themes'. The fourth and fifth steps will involve translating studies by checking first and second order concepts and themes against each other. They may be similar (reciprocal) or refutational. Step 6 is synthesising the translations to create a third order construct. Step 7 is expressing the synthesis through dissemination.

A sample of papers will be duplicate coded and discussed with the review team. This is to view the data through different perspectives (ie, a form of investigator triangulation) rather than to check for consistency in coding between reviewers, which is a more positivist approach. We will only code data that explicitly link food insufficiency with weight status, that is, we will not code experiences relating to weight status spoken in the context of wider socioeconomic

position. ZB will analyse and synthesise the entire data set. When synthesising and writing up the findings of the review, we will adhere to eMERGe Reporting Guidance, recommended to ensure complete and transparent reporting of meta-ethnographies.[51]

## Patient and public involvement

Patients and members of the public were not involved in the design of this protocol. We will consult with local community members for dissemination of the review findings.

## DISCUSSION

To the best of our knowledge, this is the first systematic review which aims to synthesise and interpret the findings of qualitative studies of food insecurity in relation to nutritional health and well-being, focusing on women and children from HICs. We believe it is important to set the study within the context of the last 12 years as, postfinancial crisis, all HICs suffered an economic crash alongside increasing poverty rates. Excluding studies with data collected prior to 2008 ensures that the synthesis is contextualised by the effects of the last financial crises and events since, such as the COVID-19 pandemic.

We recognise that for HICs outside of the USA and Canada, qualitative literature in this area is limited. In response to this we have taken a comprehensive approach to our search strategy in terms of searching stakeholder websites to locate relevant grey literature that might contain primary qualitative data. This is supplemented by screening the reference lists and citations of included studies. To maximise rigour and transparency, this qualitative systematic review was designed following established protocols, and will adhere to recommended and validated methods and reporting guidelines.

A potential limitation of the review is that it will include studies from multiple countries. It may be challenging to draw meaningful conclusions from studies involving different welfare states, social security, food aid and healthcare systems.[52 53] However, the study is set within the context of the global financial crises and not the response to this. Deciding to limit the studies to English language only might exclude some relevant studies, particularly from European countries. However, due to funding and time limitations, translation is beyond the scope of this review.

This synthesis is relevant because it is set within the context of the last financial crises and more recently, a pandemic that is beginning to result in further economic downturn. These world events have the potential to exacerbate health inequalities. It is important that research focusing on health inequalities is shaped by understanding from people who are experiencing them.[54 55] This review does that by aiming to further our understanding of the experiences of food-insecure women and children in the context of nutrition. This review will therefore provide valuable

 

insights for future research and help guide health policy and practice to support food-insecure women and children from HICs.

## ETHICS AND DISSEMINATION

The review is using secondary data, so ethical approval is not required. Review findings will be disseminated by publication in theses, peer-reviewed academic journal articles, conferences, policy and practice workshops such as those organised by Fuse: the UK Centre for Translational Research in Public Health (www.fuse.ac.uk), and circulated to the general public and stakeholder groups using social media.

**Contributors** ZB drafted the manuscript and is the guarantor of the review. All authors (ZB, NH, SS, SV, CB, JR) contributed to the development of the selection criteria. ZB, NH, SS and SV contributed to the quality appraisal assessment strategy and data extraction criteria. ZB developed the search strategy. All authors provided comments and amendments. All authors read, provided feedback and approved the final manuscript.

**Funding** This systematic review was supported by the Economic and Social Research Council (ESRC) as part of a doctoral research project.

**Disclaimer** The funder had no role in developing the systematic review protocol and will not be involved in conducting the review.

**Competing interests** None declared.

**Patient consent for publication** Not required.

**Provenance and peer review** Not commissioned; externally peer reviewed.

**ORCID iD**
Zoe Bell http://orcid.org/0000-0003-2416-4184

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
