## [Reviewer comments · BMJ Open]

ARTICLE DETAILS

TITLE (PROVISIONAL)	Food insecurity and the nutritional health and wellbeing of women and children in high-income countries: protocol for a qualitative systematic review
AUTHORS	Bell, Zoe ; Scott, Steph; Visram, Shelina; Rankin, Judith; Bambra, Clare; Heslehurst, Nicola

VERSION 1 – REVIEW

REVIEWER	Licaj, Ildir UiT - The Arctic University of Norway, Tromsø, Norway, Department of Community Medicine
REVIEW RETURNED	17-Feb-2021

GENERAL COMMENTS	This is very well detailed protocol following a gold standard methodology and a comprehensive approach and search strategy. The authors will have to deal with some important challenges during the review of the selected papers The most important one being the heterogeneity of food insecurity within and between countries and the double burden of food insufficiency and obesity. These challenges and how the authors plan to overcome them needs to be discussed with more detail. The study limits have been well discussed. The term socially acceptable needs further clarification.
--

REVIEWER	Douglas, Flora Robert Gordon University, School of Nursing and Midwifery
REVIEW RETURNED	31-Mar-2021

GENERAL COMMENTS	This is an important and timely review that will provide a contemporary picture of the existing evidence base, and the gaps and silences surrounding this urgent and critical public health problem. I accept the author's contention that this is the first review of its kind to focus on the lived experiences of the impact of living in food insecurity on the health and well-being of women and children based in high income countries. It is undoubtedly the case that this pre-existing societal vulnerability, has been significantly exacerbated by the COVID pandemic fallout, and that there is an urgent need to develop a far greater understanding about what has previously been established in this area, in order to inform policy and practice interventions that are more likely to be effective. In relation to the UK context, I believe this to be particularly important as we approach a critical juncture in the natural history of this pandemic and (hopefully) receding risk it represents to the population. During the pandemic, much public and policy focus has been placed on the operation of the food banks and food charities, which have been understandably lauded in their efforts to feed desperate people in desperate times. However, this lionisation of the food charity sector has not been
--

subject to great deal of critique in terms of their impact and efficacy in addressing the problem they purport to address, and contemporary history tells us that during periods of economic crisis, the food charity sector has grown in response to the scale of challenge, but has not tended to decline or shrink, but become entrenched in policy and public imagination as a state sponsored solution to enduring food insecurity in low income households. So, for me this is truly a moment to ask questions about those and similar types of interventions, or at the very least, within this review, to seek to see where, or if, they feature in the accounts of food insecure women and families, and establish with them directly, as intended or actual food charity recipients, their perceptions about the role those types of interventions (and others) have played in addressing or ameliorating their food insecurity experience, and its effect on their health and well-being.

The authors quite rightly indicate that a key challenge they face is interpreting the data from this review and make recommendations in such a way that is meaningful for different national and social security contexts, particularly as they also point out that they anticipate that most of the literature is likely to come from North America – which have different programmes and eligibility criteria for low income households with children to access food. So, I am curious to see what they conclude from the review in that regard.

However, the review process is very clearly laid out, with precise questions, inclusion and exclusion criteria, and an appropriate and valid process of data extraction and synthesis, that could be replicated.

One question for the authors though that I like them to consider is the following. I accept that women's particular vulnerability to food insecurity is associated with both the biological demands they experience associated with menstruation, pregnancy and lactation, and, their gender-based roles associated with their role as primary care givers of children within families. For example, it is well established that women are more economically vulnerable, globally, because they experience access constraints to the paid labour market due to their caring roles. Female-headed households, with no other economically active adult present, are the most vulnerable to chronic and more severe forms of food insecurity compared to other household types. However, whilst it is much less common for fathers to be lone parents, they do experience similar economic vulnerability, where there are no other economically active adults present, and therefore I am curious to understand why this group appears to be excluded here?

One further minor point to note is the claim in the abstract, that the outcomes the authors intend to focus on are nutrition and nutritional health, including diet, food practices, infant feeding practices, physical and mental health outcomes. Given that the authors then stress later in the body of article, that they will not consider experiences and accounts not explicitly related to food, this should be pointed out in the abstract as well - i.e. in relation to the physical and mental health outcomes they refer to in the abstract. They need to make this important distinction clearer.

One final minor point - there is a spelling error on line 48 page 9.

VERSION 1 – AUTHOR RESPONSE

Reviewer 1:

Minor comments

- Thank you for this feedback

The authors will have to deal with some important challenges during the review of the selected papers. The most important one being the heterogeneity of food insecurity within and between countries and the double burden of food insufficiency and obesity. These challenges and how the authors plan to overcome them needs to be discussed with more detail.

- In response to the comment on heterogeneity:

- o We have already set the study within the context of the global financial crises from 2008 onwards.

This period was chosen as OECD high-income countries all experienced rising poverty rates because of the financial crises.

- o Since originally drafting the protocol, the review has narrowed to include only high-income countries within a European context. This was due to a high volume of papers through to full-text screening (n= 258) which was beyond the project's timing and funding limits. Deviance from the protocol will be reported in the follow-on manuscript of the review, or if preferred, the protocol can be amended.

- o During the data extraction phase, we will actively look for reporting on each study's context and explore these issues in the analysis.

- o We have revised the study selection section of the manuscript, pg. 9: "During this stage we will actively look for reporting on each study's context to explore in the analysis."

- In response to the comment on the double burden of food insecurity:

- o Thank you, for your comment. We recognise the double burden of food insufficiency (inadequate nutrition) and obesity in high-income countries, and their co-existence because they are both consequences of socio-economic disadvantage. From a life course perspective, food insecurity during the first 1000 days of life and in childhood may have longer term nutritional health and wellbeing impacts, including obesity. Exploring women of childbearing age, pregnant women's, mothers, and children's experiences of food insecurity might highlight a coping strategy that represents a potential mechanism by which food insufficiency could be related to the risk of developing obesity. To address this issue, we are focusing explicitly on experiences related to nutrition.

- o During the data coding stage, in relation to this aspect, we will only code data that explicitly links food insufficiency with weight status i.e., experiences relating to weight status spoken in the context of wider socioeconomic position will not be coded.

- o We have revised the data synthesis section of the manuscript pg.9: "We will only code data that explicitly links food insufficiency with weight status i.e., we will not code experiences relating to weight status spoken in the context of wider socioeconomic position."

The study limits have been well discussed.

- Thank you for this feedback

The term socially acceptable needs further clarification...

- We have expanded on this terminology within the manuscript to clarify its meaning

- o We have added this sentence to the introduction section of the manuscript pg. 3 "Accessing food in a socially acceptable way means people don't have to live off 'free' food re-distributed by charity."

Reviewer 2:

Minor comments

Full comments:

- Thank you for this feedback, we agree this is a timely and important topic for review

During the pandemic, much public and policy focus has been placed on the operation of the food banks and food charities, which have been understandably lauded in their efforts to feed desperate people in desperate times. However, this lionisation of the food charity sector has not been subject to great deal of critique in terms of their impact and efficacy in addressing the problem they purport to address, and contemporary history tells us that during periods of economic crisis, the food charity sector has grown in response to the scale of challenge, but has not tended to decline or shrink, but become entrenched in policy and public imagination as a state sponsored solution to enduring food insecurity in low income households. So, for me this is truly a moment to ask questions about those and similar types of interventions, or at the very least, within this review, to seek to see where, or if, they feature in the accounts of food insecure women and families, and establish with them directly, as intended or actual food charity recipients, their perceptions about the role those types of interventions (and others) have played in addressing or ameliorating their food insecurity experience, and its effect on their health and well-being.

- Thank you for this suggestion. We agree that this is an important topic. Our review is focussing on the perspectives of food insecurity on diet and nutrition, and we are using a data driven approach to coding the data and developing themes in the meta-ethnography. How much we can discuss this important topic in this review will depend on whether it is raised by the participants in the included studies. If this features in the findings, then it will be reported in the follow-on manuscript of the review.

The authors quite rightly indicate that a key challenge they face is interpreting the data from this review and make recommendations in such a way that is meaningful for different national and social security contexts, particularly as they also point out that they anticipate that most of the literature is likely to come from North America – which have different programmes and eligibility criteria for low income households with children to access food. So, I am curious to see what they conclude from the review in that regard.

- Thank you for this feedback.

- o Since originally drafting the protocol, the review has narrowed to include only high-income countries within a European context. This was due to a high volume of papers through to full-text screening (n= 258) which was beyond the project's timing and funding limits. Deviance from the protocol will be reported in the follow-on manuscript of the review, or if preferred, the protocol can be amended.

- o During the data extraction phase, we will actively look for reporting on each study's context and explore these issues in the analysis.

- o We have revised the study selection section of the manuscript, pg. 9: "During this stage we will actively look for reporting on each study's context to explore in the analysis."

However, the review process is very clearly laid out, with precise questions, inclusion and exclusion criteria, and an appropriate and valid process of data extraction and synthesis, that could be replicated.

- Thank you for these comments relating to the rigorous methods we are using

One question for the authors though that I like them to consider is the following. I accept that women's particular vulnerability to food insecurity is associated with both the biological demands they experience associated with menstruation, pregnancy and lactation, and, their gender-based roles associated with their role as primary care givers of children within families. For example, it is well established that women are more economically vulnerable, globally, because they experience access constraints to the paid labour market due to their caring roles. Female-headed households, with no other economically active adult present, are the most vulnerable to chronic and more severe forms of food insecurity compared to other household types. However, whilst it is much less common for fathers to be lone parents, they do experience similar economic vulnerability, where there are no

other economically active adults present, and therefore I am curious to understand why this group appears to be excluded here?

- Fathers and other caregivers are included in the population group answering the review's second aim exploring food insecure household's accounts of their children's nutritional health and wellbeing. Although links are emerging to understand how a father's diet around the time of conception impacts his child, this evidence base is still limited. Animal studies have shown both direct (sperm quality, epigenetic status, DNA integrity) and indirect (seminal fluid composition) mechanisms through which a father can affect their child's health through fetal development mechanisms.
- However, this review is part of a wider PhD project which looks at the social determinants of food insecurity and its nutritional impact amongst women and children, focusing on the first 1000 days of life. There is increasing evidence linking women's preconception and pregnancy nutrition and the impact on their child's health through fetal development mechanisms. Further, the Developmental Origins of Health and Disease concept suggests that poor developmental experience can increase susceptibility of non-communicable diseases in later life. The first 1000 days of life (pregnancy to 2 years) is a critical developmental window. Food insecurity during this period has a double burden on both a woman's nutritional status, and her unborn child's nutritional health and wellbeing and susceptibility to disease in later life, contributing to health inequalities before a baby is born.
- We have added a sentence to contextualise the review within a wider PhD project
 - o See introduction section of the manuscript pg. 4: "this review is part of a wider PhD research programme which looks at the social determinants of food insecurity and its nutritional impact amongst women and children, focusing on the first 1000 days of life".

One further minor point to note is the claim in the abstract, that the outcomes the authors intend to focus on are nutrition and nutritional health, including diet, food practices, infant feeding practices, physical and mental health outcomes. Given that the authors then stress later in the body of article, that they will not consider experiences and accounts not explicitly related to food, this should be pointed out in the abstract as well - i.e., in relation to the physical and mental health outcomes they refer to in the abstract. They need to make this important distinction clearer.

- Thank you for your suggestion. We have now amended the abstract narrative:
- We have changed this sentence in the abstract of the manuscript, pg.2: "This review aims to understand experiences of food insecurity and how it affects women and children's nutritional health and wellbeing, focusing on experiences explicitly related to nutrition."

One final minor point - there is a spelling error on line 48 page 9.

- We have amended this error